# Prone Positioning: A Safe and Effective Procedure in Pregnant Women Presenting with Severe Acute Respiratory Distress Syndrome

**DOI:** 10.3390/vaccines10122182

**Published:** 2022-12-19

**Authors:** Gilmar de Souza Osmundo, Cristiane de Freitas Paganotti, Rafaela Alkmin da Costa, Thiago Henrique dos Santos Silva, Paula Carolina Bombonati, Luiz Marcelo Sa Malbouisson, Rossana Pulcineli Vieira Francisco

**Affiliations:** 1Disciplina de Obstetricia, Departamento de Obstetricia e Ginecologia, Faculdade de Medicina FMUSP, Universidade de Sao Paulo, Sao Paulo 05403-900, Brazil; 2Divisao de Anestesiologia, Hospital das Clinicas HCFMUSP, Faculdade de Medicina, Universidade de Sao Paulo, Sao Paulo 05403-900, Brazil

**Keywords:** COVID-19, respiratory distress syndrome, prone position, pregnancy, intensive care unit

## Abstract

Prone positioning (PP) improves oxygenation and survival in patients with severe acute respiratory distress syndrome (ARDS). Data regarding feasibility and effectiveness of PP in pregnancy are lacking. This subgroup analysis of a cohort study that included mechanically ventilated pregnant women presenting with severe acute respiratory syndrome coronavirus 2 (SARS-CoV-2)-induced ARDS who underwent PP aims to assess the efficacy and safety of PP. Ventilatory and gasometric parameters were evaluated at baseline (T_0_) and in prone (T_1_) and supine (T_2_) positions. Obstetric outcomes were also assessed. Sixteen cases at an average of 27.0 (22.0–31.1) gestational weeks of pregnancy were included. Obesity and hypertension were frequent comorbidities. PP was associated with a >20% increase in PaO_2_ levels and in PaO_2_/FiO_2_ ratios in 50% and 100% of cases, respectively. The PaO_2_/FiO_2_ ratio increased 76.7% (20.5–292.4%) at T_1_ and 76.9% (0–182.7%) at T_2_. PP produced sustained improvements in mean PaO_2_/FiO_2_ ratio (*p* < 0.001) and PaCO_2_ level (*p* = 0.028). There were no cases of emergency delivery or suspected fetal distress in pregnancies ≥25 weeks during the 24 h period following PP. PP is safe and feasible during pregnancy, improving PaO_2_/FiO_2_ ratios and helping to delay preterm delivery in severe ARDS.

## 1. Introduction

Prone positioning (PP) has been evidenced to be a useful strategy in patients with severe acute respiratory distress syndrome (ARDS), decreasing mortality and improving oxygenation during mechanical ventilation. PP is indicated in those in whom the ratio of the partial pressure of oxygen to the fraction of inspired oxygen (PaO_2_/FiO_2_ ratio) is less than 150 mmHg [1,2].

Physiological changes make managing ARDS episodes challenging in pregnant women. Oxygen consumption is increased during pregnancy, demanding oxygen saturation (SaO_2_) > 95% and arterial partial pressure of oxygen (PaO_2_) > 70 mmHg to ensure adequate fetal oxygenation [3]. At a late stage of pregnancy, the gravid uterus induces cephalic displacement of the diaphragm, leading to atelectasis of the pulmonary lower lobes, particularly in sedated and mechanically ventilated patients [4,5]. Moreover, the weight of mediastinal structures may impose an additional load on the lower lobes, further worsening the loss of aeration [6]. In such patients, PP may be a valuable resource for treating ARDS by relieving mechanical factors associated with the collapse of lung regions.

Pregnant women are often excluded from trials investigating the use of PP despite its potential benefits for blood oxygenation [3,7,8,9]; thus, literature on this topic is scarce. Due to special concerns regarding overloading the uterus, aortocaval compression, and fetal wellbeing monitoring [8], most caregiver teams fear use of PP in women with large gravid uteri.

In face of the recent burden of severe acute respiratory syndrome coronavirus 2 (SARS-CoV-2) on the pregnant population, confidence and proficiency in offering and managing PP in pregnant women may provide a low-cost, available, safe, and effective intervention to those who are hypoxemic and even under mechanical ventilation. Recently, technical guidelines for PP have been published, focusing on pregnant women [10,11]. This study aims to describe the feasibility, safety, and effectiveness of PP in a series of pregnant women with ARDS admitted to the intensive care unit (ICU) of a tertiary teaching hospital.

## 2. Materials and Methods

This is a secondary retrospective analysis of an ongoing Brazilian cohort study titled “Exploratory study on coronavirus disease (COVID-19) in pregnant women”. The present sub-analysis aims to evaluate clinical and obstetric outcomes following PP in pregnant women with severe acute respiratory syndrome coronavirus 2 (SARS-CoV-2)-induced ARDS who were admitted to the ICU of a tertiary teaching hospital (Hospital das Clínicas, Faculdade de Medicina da Universidade de São Paulo, São Paulo, Brazil). Data of women enrolled in the study between April 2020 and August 2021 were obtained.

Patients were selected according to the following inclusion criteria: singleton pregnancy, gestational age at PP ≥ 20 weeks, diagnosis of SARS-CoV-2 confirmed via real-time polymerase chain reaction (RT-PCR) from samples obtained from the respiratory tract (nasopharyngeal or tracheal) after the third day of symptom onset, PaO_2_/FiO_2_ ratio < 150 mmHg, need for mechanical ventilation, and minimum PP duration of 12 h.

Patients with severe ARDS who were on mechanical ventilation and received sedation and paralysis were ventilated with a tidal volume of 6 mL/kg of ideal weight as a routine of the ICU, aiming to achieve a driving pressure equal to or lower than 15 mmHg.

The prone position procedure was performed according to a standardized local protocol, and a multidisciplinary team consisting of physicians, nurses, and physical therapists performed the positioning. The orotracheal tube, catheters, and patient’s head were stabilized while switching positions. The bone prominences and nipples were protected, and pads were placed under the chest, pelvis, and knees to avoid compression of the gravid maternal abdomen during PP. The procedure followed a safety checklist. During PP, the head and upper limbs (swimmer’s position) were switched every 2 h to prevent pressure injury. After PP, blood gas analysis was performed to evaluate the initial PaO/FiO_2_ ratio response.

Fetal surveillance consisted of computerized cardiotocography, fetal biophysical profiling, and umbilical artery Doppler. These parameters were assessed immediately before PP and soon after returning to the supine position in cases with a gestational age ≥25 weeks. Cardiotocography was only performed during PP if a maternal status worsening was expected.

Data regarding maternal demographic characteristics and clinical and obstetric outcomes were assessed. Respiratory parameters considered included the following: PaO_2_, PaO_2_/FiO_2_ ratio, PaCO_2_, tidal volume, and positive end-expiratory pressure (PEEP). Gas blood analysis was performed within 12 h of prone- and supine-position placement. Clinical and gasometric parameters were considered at the following three time points: T_0_ (baseline, at the time of PP indication), T_1_ (within the first 12 h after PP), and T_2_ (within the first 12 h after returning to the supine position) after the first prone position of each subject.

### Statistical Analyses

The categorical variables are described as absolute and relative frequencies. The continuous variables are described as mean and standard deviation (SD) or median, with minimum and maximum. The continuous variables were compared using one-way analysis of variance (ANOVA), and pairwise comparisons were performed using Tukey’s post hoc test. Cohen’s f calculation was applied to variables presenting *p*-value < 0.05 in ANOVA test to estimate the effect size. Categorical variables were compared using the chi-square test. Kaplan–Meier survival analysis was performed to assess the duration women remained pregnant after placement in the prone position. Statistical significance was set at two-tailed *p*-value < 0.05. Statistical analysis was performed using Statistical Package for the Social Sciences (SPSS) software (SPSS Statistics for Windows, version 21, IBM, Chicago, IL, USA).

## 3. Results

During the study period, 139 pregnant women presenting with SARS-CoV-2-induced ARDS were referred to our ICU. Among them, 18 patients underwent PP. Two subjects did not meet inclusion criteria; therefore, they were excluded from the analysis. One patient was excluded due to a duration of PP < 2 h due to hemodynamic instability, and another was excluded due to presentation with mild ARDS that did not require mechanical ventilation. Thus, 16 pregnant women were included in the study.

Patients were aged 31.5 (22.0–46.0) years and had a body mass index of 36.0 (23.4–47.9) kg/m^2^. Obesity and systemic hypertension were the most prevalent comorbidities, accounting for 73.3% and 40.0% of cases, respectively. Most women were multiparous (87.5%), and their gestational age at the first PP was 27.0 (22.0–31.1) weeks. Nine patients (56.3%) underwent PP within the first 24 h after referral to our service (Table 1).

At the indication of PP, patients were on mechanical ventilation with tidal volume of 6.1 (±0.8) ml/kg, PEEP of 12.6 (±3.2) cmH_2_O, and had a FiO_2_ value of 1.0 in 56.3% of cases. Patients were administered a sedative–analgesic regimen plus a neuromuscular blocking agent. Furthermore, 50% of patients required norepinephrine. The most frequently used sedatives were midazolam (78.6%) and propofol (64.3%). Further, all patients received fentanyl and cisatracurium. Baseline blood gas parameters at the indication of PP were as follows: PaO_2_ of 85.1 ± 15.8 mmHg, PaCO_2_ of 48.3 ±10.4 mmHg, and PaO_2_/FiO_2_ ratio of 102.7 ± 26.9 mmHg.

Patients were maintained in the prone position for 20.0 (14.0–24.0) h and the number of PP events was 1.5 (1.0–7.0) per patient. After returning to the supine position, a >20% increase in PaO_2_ and PaO_2_/FiO_2_ ratio was observed in 56.3% and 81.3% of patients, respectively. The observed increment in the PaO_2_/FiO_2_ ratio was 76.7% (20.5–292.4%) in the prone position and 76.9% (0–182.7%) in the supine position (Table 2).

Regarding obstetric outcomes, 10 patients had an emergency C-section delivery during the ICU stay at 29.8 (27.1–31.7) weeks of gestation and had a birth weight of 1484 (985–2010) g. Deliveries occurred 7 (2–24) d after the first positioning and 5 (2–4) d after the last PP. The main indication for delivery was fetal distress (*n* = 7), consisting of refractory oligohydramnios (*n* = 3), non-reassuring fetal rate (*n* = 2), and a fetal biophysical profile < 6 (*n* = 2).

Five patients remained pregnant at hospital discharge and had uneventful deliveries between 37 and 39 weeks of gestation. Figure 1 shows a Kaplan–Meier curve depicting the percentage of women remaining pregnant within the first 30 days following PP. There was one case of stillbirth during PP in a patient presenting with severe baseline ARDS and respiratory acidosis (arterial pH, 7.21; PaO_2_, 89.5 mmHg; PaO_2_/FiO_2_ ratio, 89.5 mmHg; and PaCO2, 55.6 mmHg) at the 24th week of gestation.

One-way ANOVA was performed to compare effects of PP on blood gas parameters. Prone positioning was associated with the sustained improvement of PaO_2_/FiO_2_ ratio (T_0_: 102.7 ± 26.9 mmHg, T_1_: 189.2 ± 71.5 mmHg and T_2_: 187.1 ± 71.6 mmHg, *p* < 0.001) and PaCO_2_ (T_0_: 48.3 ± 10.4 mmHg, T_1_: 44.4 ± 10.0 mmHg and T_2_: 38.7 ± 8.8 mmHg, *p* = 0.028; Table 3). PP had a medium effect size for the sustained improvement of the PaO_2_/FiO_2_ ratio (Cohen’s f = 0.56) and PaCO_2_ (Cohen’s f = 0.46). Post hoc analyses confirmed that PP was related to an increased PaO_2_/FiO_2_ ratio at T_1_ (*p* = 0.001) and T_2_ (*p* = 0.001), and to a reduced PaCO_2_ ratio at T_2_ (*p* = 0.022), when compared to baseline (Figure 2).

There was no difference in PaCO_2_ or PaO_2_/FiO_2_ ratio observed when prone and supine positions were compared. There was no association between PP and a decrease in mean arterial pressure (MAP, *p* = 0.986) or an increase in the prevalence of norepinephrine administration (*p* = 0.933). There were no cases of ventilation-associated pneumonia or pulmonary thromboembolism over two weeks after PP.

## 4. Discussion

To date, this is the largest study investigating cases involving PP in pregnant patients with ARDS. We verified that PP is a safe and effective procedure in women on mechanical ventilation due to severe ARDS who were in the second- and third-trimester of pregnancy. Furthermore, we achieved favorable obstetric outcomes in the very critically ill group of patients. Nearly one-third of our patients left the ICU without delivering.

The literature regarding PP during pregnancy is scarce and comprises reports of PP feasibility and successful cases [2,3,7,8,9,12,13,14]. In our study, in addition to verifying the feasibility of a PP protocol in pregnant women with moderate to severe ARDS, we assessed the effectiveness of the procedure by verifying significant sustained improvements in PaO_2_/FiO_2_ ratio and PCO_2_ values after PP. These favorable outcomes may have contributed to delayed deliveries in severe ARDS patients undergoing PP.

There are multiple mechanisms by which the prone position contributes to improved oxygenation including increasing the homogeneity of ventilation and perfusion of the entire lung, relieving alveolar compression in the dorsal lung regions, and changing chest compliance, which improves the distribution of gases toward the ventral and para-diaphragmatic lungs. The impact of PP persists after the patient returns to a supine position [2,15].

A PaO_2_/FiO_2_ ratio < 150 mmHg is a classical indication for PP. During and after PP, 62.5% and 68.7% of patients presented with PaO_2_/FiO_2_ > 150 mmHg, which is a treatment target when managing patients with ARDS. From another point of view, an increase greater than 20% in PaO_2_/FiO_2_ ratio indicates a patient is a responder to PP [16]. All patients in the present study were classified as responders during PP, and half of them sustained this response after placement in the supine position. Criteria indicating responders or non-responders are not clearly related to mortality improvement, although literature findings on the subject are controversial [1,17,18,19,20]. Nonetheless, improvements may have contributed to the delayed preterm delivery in severe ARDS observed in this study.

Maintaining low levels of PaCO_2_ facilitates gas exchange between the mother and fetus and is important for avoiding fetal distress during the treatment of pregnant women with ARDS. Carbon dioxide elimination is an additional benefit of the prone position because recruitment and perfusion of collapsed lung units reduce pulmonary shunting and decrease PaCO_2_ [15]. After PP, we observed a significant decrease in PaCO_2_, which is an additional target when managing ARDS during pregnancy.

In this study, PP discontinuation was required in one patient due to hemodynamic instability that occurred less than 2 h after PP; therefore, the patient was not included in the analysis. In general, PP did not affect MAP or increase the requirement for vasopressors. Overall, PP was well tolerated and was administered for approximately 20 h.

Hemodynamic instability is not an absolute contraindication for PP. In fact, half of the patients included in the present study required vasopressors. In the PROSEVA study, 72% of patients who underwent PP received vasopressors, and the procedure did not induce hemodynamic side effects and improved cardiovascular parameters [1,15].

This study comprises a very small sample size. However, to the best of our knowledge, this is the largest series of cases to address the effectiveness of PP in improving oxygenation parameters during ARDS in pregnancy. Although our sample size was not large, it was sufficient for revealing clear improvement in PO_2_/FiO_2_ ratio after PP. Thus, this study shows a significant effect of PP on ARDS during pregnancy. The small sample size, observational nature of the study, and lack of a control group prevented us from being able to infer that PP was decisive in the clinical improvement of pregnant subjects. Larger and more specifically designed studies are required for these purposes.

In the population assessed, there were 10 cases of preterm delivery during the ICU stay, and only 1 of them was due to a worsening maternal status. In a multicenter cohort study that enrolled patients from 12 hospitals in the USA, Pierce-Williams et al. described a subgroup of 20 critically ill pregnant women (19 needed intubation and 4 underwent PP), among which 17 delivered during hospitalization (13 due to maternal status) [21]. One-third of our sample delivered at term.

The main indication for delivery in our study was fetal distress. Remarkably, there were no cases of emergency cesarean section or suspected fetal distress throughout PP or in the first 24 h after returning to the supine position. Deliveries during the ICU stay occurred after 7 (2–24) d of PP and at least 2 d after the patient last underwent PP. Fetal well-being is a frequent concern when treating pregnant women in the ICU; therefore, it might be seen as an impediment to PP. However, our findings highlight the safety of PP in pregnant women, confirming results obtained by Oliveira et al. [7].

Our study population presented substantial prevalence of obesity and systemic hypertension, which are known risk factors for severe forms of SARS-CoV-2 [22,23]. Previously, Gomez et al. [24] evaluated all cases of SARS-CoV-2 among pregnant women from our institution, finding that more severe cases had larger body mass index and higher prevalence of systemic hypertension. Further, authors described that most severe cases were associated with preterm birth, fetal distress, and maternal and neonatal death as compared to mild and moderate cases [24].

Our study suggests that PP is a valid tool for managing pregnant women with severe ARDS. PP did not prevent preterm delivery in 62.5% of our population; nonetheless, it was able to delay delivery. Considering the care of pregnant women, delaying preterm delivery even for a few days allows for additional fetal lung maturation and delivery at a greater gestational age, providing a potential neonatal benefit. Moreover, PP improved maternal respiratory and metabolic parameters. As a result, PP allowed patients to undergo surgical procedures in less critical conditions, providing a potential maternal benefit due to surgical risk reduction. Notably, PP was indicated after a median duration of 1 d after mechanical ventilation was initiated, suggesting that ventilation may be considered an early indication for PP. Prior evidence has shown that early initiation of PP improves the post-PP response [15], which could be one explanation for our results showing that it ameliorates hypoxia in pregnant women.

## 5. Conclusions

Prone positioning may be safely used during pregnancy, even after the uterus is enlarged throughout second and third trimesters. PP seems to be effective in improving the PaO_2_/FiO_2_ ratio and PCO_2_ level and may help delay preterm deliveries. Additional studies will be needed to better assess middle- and long-term outcomes of PP in pregnancy as well as to establish specific protocols and goals related to the use of PP to treat the pregnant population.

## Figures and Tables

**Figure 1 vaccines-10-02182-f001:**
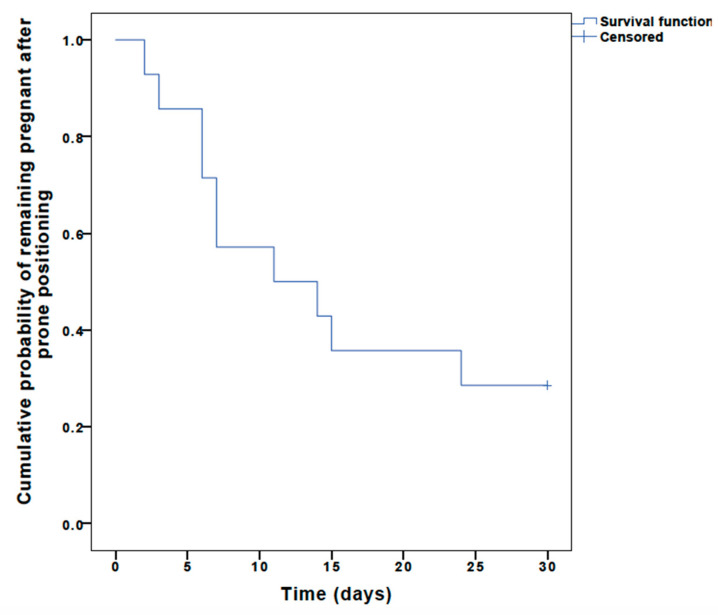
Kaplan–Meier survival curve representing time that women remained pregnant after prone positioning.

**Figure 2 vaccines-10-02182-f002:**
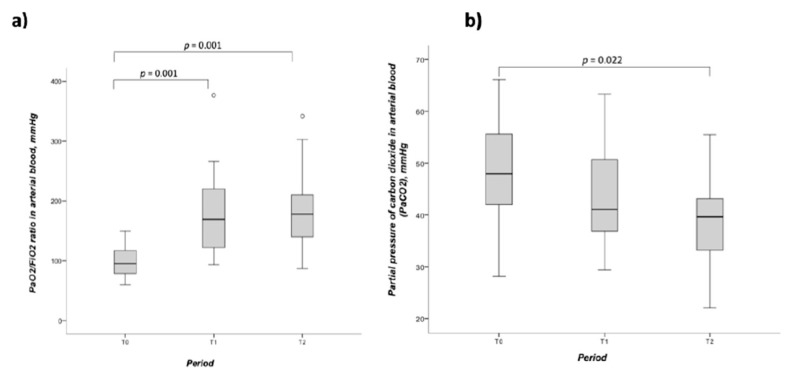
Arterial blood gas measures at baseline (T0), prone position (T1), and supine position (T2) in pregnant women with SARS-CoV-2 induced ARDS: (**a**) PaO_2_/FiO_2_ ratio; (**b**) PaCO_2_. Abbreviations: paCO_2_: partial pressure of carbon dioxide in arterial blood; paO_2_: partial pressure of oxygen in arterial blood; FiO_2_: inspired oxygen fraction. (Tukey’s post-hoc test).

**Table 1 vaccines-10-02182-t001:** Demographic, clinical, and obstetric parameters of pregnant women undergoing prone positioning (*n* = 16).

Parameters	Value
Maternal age, years, median (min–max)	31.5 (22.0–46.0)
Body mass index, kg/m^2^, median (min–max)	36.0 (23.4–47.9)
Ethnicity *	
Caucasian, N (%)	9 (64.3%)
Non-Caucasian, N (%)	5 (35.7%)
Obesity, N (%)	11 (73.3%)
Systemic hypertension, N (%)	6 (40.0%)
Gestational diabetes, N (%)	2 (13.3%)
Asthma, N (%)	1 (6.7%)
GA at first prone positioning, weeks, median (min–max)	27.0 (22.0–31.1)
Duration of symptoms before ICU admission, days, median (min–max)	8.0 (2.0–14.0)
Duration of symptoms before prone positioning, days, median (min–max)	11 (5.0–18.0)
Duration of ICU stay before prone positioning, days, median (min–max)	1.0 (0–8.0)
TMV before prone positioning, days, median (min–max)	1.5 (0–6.0)

GA: gestational age; ICU: intensive care unit; TMV: time on mechanical ventilation. *: data unavailable for 2 patients.

**Table 2 vaccines-10-02182-t002:** Maternal and perinatal outcomes following prone positioning.

Parameters	Measured Value
Time on prone positioning, h, median (min–max)	20.0 (14.0–24.0)
Number of prone positionings, median (min–max)	1.5 (1.0–7.0)
Increase in PaO_2_ > 20% (T_1_), N (%)	8 (50%)
Increase in PaO_2_ > 20% (T_2_), N (%)	9 (56.3%)
Increase in PaO_2_/FiO_2_ > 20% (T_1_), N (%)	16 (100%)
Increase in PaO_2_/FiO_2_ > 20% (T_2_), N (%)	13 (81.3%)
PaO_2_/FiO_2_ response (T_1_), %, median (min–max)	76.7 (20.5–292.4)
PaO_2_/FiO_2_ response (T_2_), %, median (min–max)	76.9 (0–182.7%)
Time on mechanical ventilation, days, median (min–max)	10.5 (5.0–28.0)
Time in UCI, d, median (min–max)	19.5 (9.0–36.0)
Venous–venous ECMO, N (%)	1 (6.2%)
Delivery during ICU admission, N (%) *	10 (62.5%)
Time of first prone-delivery**, days, median (min–max)	7.0 (2.0–24.0)
Time of last prone-delivery**, days, median (min–max)	5 (2–4)
GA at delivery, weeks, median (min–max) **	29.8 (27.1–31.7)
Birth weight, grams, median (min–max) **	1484 (985–2010)
Indications for delivery	
Fetal distress, N (%)	7 (70%)
Maternal complications, N (%)	1 (10%)
Preterm labor, N (%)	1 (10%)
PPROM, N (%)	1 (10%)
Maternal death, N (%)	2 (12.5%)
Stillbirth, N (%)	1 (6.2%)

*: one patient had stillbirth and induced vaginal delivery at 24 weeks of gestation. **: only for those who delivered during ICU admission (*n* = 10). ECMO: extracorporeal membrane oxygenation; GA: gestational age; PPROM: preterm premature rupture of membranes; T_1_: prone position; T_2_: supine position.

**Table 3 vaccines-10-02182-t003:** Respiratory and hemodynamic responses to prone positioning (*n* = 16).

Parameters	T_0_	T_1_	T_2_	*p*
MAP, mmHg, mean (±SD)	85.9 (±11.6)	85.3 (±10.3)	85.7 (±12.2)	0.986
Norepinephrine, N (%)	8 (50%)	10 (62.5%)	9 (56.3%)	0.933
PaO_2_, mmHg, mean (±SD)	85.1 (±15.8)	103.7 (±25.4)	102.6 (±25.4)	0.037 ^1^
PaO_2_/FIO_2_ ratio, mmHg, mean (±SD)	102.7 (±26.9)	189.2 (±71.5)	187.1 (±71.6)	<0.001 ^2^
PaCO_2_, mmHg, mean (±SD)	48.3 (±10.4)	44.4 (±10.0)	38.7 (±8.8)	0.028 ^3^
Tidal volume (ml/kg), mean (±SD)	6.1 (±0.8)	6.0 (±0.8)	6.0 (±0.7)	0.979
PEEP, cmH2O, mean (±SD)	12.6 (±3.2)	12.2 (±3.0)	12.6 (±2.4)	0.910
PaO_2_/FIO_2_ > 150 mmHg, N (%)	0	10 (62.5%)	11 (68.6%)	<0.001

MAP: mean arterial pressure; PaCO_2_: partial pressure of carbon dioxide in arterial blood; PaO_2_: partial pressure of oxygen in arterial blood; PEEP: positive end expiratory pressure; SD: standard deviation; T_0_: baseline; T_1_: prone position; T_2_: supine position. ^1^ Cohen’s f = 0.36; ^2^ Cohen’s f = 0.56; ^3^ Cohen’s f = 0.46.

## Data Availability

The data supporting the findings of this study are available from the corresponding author [GSOJ] upon reasonable request.

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
