# Peer review of "Prone Positioning: A Safe and Effective Procedure in Pregnant Women Presenting with Severe Acute Respiratory Distress Syndrome"

_vaccines, 2022, doi:10.3390/vaccines10122182_

Round 1

Reviewer 1 Report

This is a useful and well researched and presented study.

Please emphasise in your Abstract, Results and Discusion that this is a tiny sample of  patients and that 11 were obese and  6 hypertensive.

Can you present data on outcomes for the otehr pregnant patients admitted who did not receive PP. This would be helpful.

Author Response

Reviewer 1

We thank your comments. We have replied all the comments and made the appropriate suggested modifications, which are highlighted in the text.

We hope you will find the paper suitable for publication in the Vaccines.

With best wishes,

  1. Please emphasize in your Abstract, Results and Discussion that this is a tiny sample of patients and that 11 were obese and 6 hypertensive.

We agree with your comments that this is a very small sample size with a high prevalence of systemic hypertension and obesity. Thus, we mentioned that “Obesity and hypertension were frequent comorbidities” in the abstract.

We also included a further discussion about those comorbidities: “Our study population presented substantial prevalence of obesity and systemic hypertension, which are known risk factors for severe forms of SARS-CoV-2. [22,23]”.

We made sure to emphasize the small sample size in the discussion: “This study comprises a very small sample size. However, to the best of our knowledge, this is the largest series of cases to address the effectiveness of PP in improving oxygenation parameters during ARDS in pregnancy”.

  1. Can you present data on outcomes for the other pregnant patients admitted who did not receive PP. This would be helpful.

We appreciate your comment regarding outcomes from patients who did not undergo prone positioning (PP). In our study, we aimed to focus on the feasibility and safety of PP in pregnant women. However, our group has already published a paper analyzing maternal and neonatal outcomes according to the severity of SARS-CoV-2. Severe cases were associated to fetal and maternal death, preterm birth, and admission to neonatal intensive care unit (Gomez, 2022).

Thus, we included the following sentence in the discussion: Previously, Gomez et al. [24] evaluated all cases of SARS-CoV-2 among pregnant women from our institution, finding that more severe cases had larger body mass index and higher prevalence of systemic hypertension. Further, authors described that most severe cases were associated to preterm birth, fetal distress, and maternal and neonatal death as compared to mild and moderate cases. [24]”.

Reviewer 2 Report

Dear Authors,

I have read with interest the manuscript and I think that it is very interestig and very well written.

I have only few questions for you

1) Have you data related this evaluation in non Covid pregnant?

2) Please add a control group of pregnant patient with ARDS and Covid that did not perform the PP

3) Please add the power calculation.

Author Response

Reviewer 2

We thank your comments. We have replied all the comments and made the appropriate suggested modifications, which are highlighted in the text.

We hope you will find the paper suitable for publication in the Vaccines.

With best wishes,

  1. Have you data related this evaluation in non Covid pregnant?

We would like to acknowledge for your comments and revision to our manuscript. Before SARS-CoV-2 pandemics, prone positioning (PP) as a treatment option for pregnant women presenting ARDS was a controversial issue due to the lack of evidence for safety in such a population. On the other hand, ARDS is a very infrequent illness among pregnant women. Thus, unfortunately, we do not have data regarding PP in non-covid ARDS pregnant women.

However, in a previous study carried out in our service, in which the prone position was tested in women with low-risk pregnancies in relation to possible changes in maternal hemodynamic parameters and also in fetal well-being, we observed that the prone position was safe for pregnant women and your fetuses (Oliveira, 2017 / reference [7]).

  1. Please add a control group of pregnant patient with ARDS and Covid that did not perform the PP

We agree that a case-control analysis would be a great strategy to assess efficacy of PP. However, this is a retrospective observational analysis of patients presenting a severe and life-threatening condition. Our service opted for performing PP in all patients with clinical indication of such procedure. Unfortunately, we do not have a matching control group because patients who did not undergo PP had a less severe disease.

On the other hand, our group has already published a paper analyzing maternal and neonatal outcomes according to the severity of SARS-CoV-2. Severe cases were associated to fetal and maternal death, preterm birth, and admission to neonatal intensive care unit (Gomez 2022).

Thus, we included the following sentence in the discussion: “Previously, Gomez et al. [24] evaluated all cases of SARS-CoV-2 among pregnant women from our institution, finding that more severe cases had larger body mass index and higher prevalence of systemic hypertension. Further, authors described that most severe cases were associated to preterm birth, fetal distress, and maternal and neonatal death as compared to mild and moderate cases. [24]”.

3.Please add the power calculation

To estimate power calculation and effect size, we calculated the Cohen’s f value for those variables presenting statistically significant differences in the ANOVA test.

PP had a medium effect size for the sustained improvement of PaO2/FiO2ratio (Cohen’s f = 0.56) and PaCO2 (Cohen’s f = 0.46). We included those values and the disclosure of the medium size effect in the results and in table 3.

Round 2

Reviewer 1 Report

The authors have appropriately stressed that this is a tiny sample. This thus sets the context that this is a research note rather than a full article.

Reviewer 2 Report

Dear Authors,

I have read the manuscript and I think that it has been improved.

I have not further comments